# FoveaBox: Beyound Anchor-based Object Detection

## Abstract

We present FoveaBox, an accurate, flexible, and completely anchor-free framework for object detection. While almost all state-of-the-art object detectors utilize predefined anchors to enumerate possible locations, scales and aspect ratios for the search of the objects, their performance and generalization ability are also limited to the design of anchors. Instead, FoveaBox directly learns the object existing possibility and the bounding box coordinates without anchor reference. This is achieved by: (a) predicting category-sensitive semantic maps for the object existing possibility, and (b) producing category-agnostic bounding box for each position that potentially contains an object. The scales of target boxes are naturally associated with feature pyramid representations. We demonstrate its effectiveness on standard benchmarks and report extensive experimental analysis. Without bells and whistles, FoveaBox achieves state-of-the-art single model performance on the standard COCO detection benchmark. More importantly, FoveaBox avoids all computation and hyper-parameters related to anchor boxes, which are often sensitive to the final detection performance. We believe the simple and effective approach will serve as a solid baseline and help ease future research for object detection.

## 1 Introduction

Object detection requires the solution of two main tasks: *recognition* and *localization*. Given an arbitrary image, an object detection system needs to determine whether there are any instances of semantic objects from predefined categories and, if present, to return the spatial location and extent. To add the localization functionality to generic object detection systems, sliding window approaches have been the method of choice for many years (Lampert et al., 2008; Felzenszwalb et al., 2010; Liu et al., 2018).

Recently, deep learning techniques have emerged as powerful methods for learning feature representations automatically from data (Simonyan & Zisserman, 2014; He et al., 2016; Huang et al., 2017a). For object detection, the anchor-based Region Proposal Networks (Ren et al., 2015) are widely used to serve as a common component for searching possible regions of interest for modern object detection frameworks (Liu et al., 2016; He et al., 2017; Lin et al., 2018). In short, anchor method suggests dividing the box space into discrete bins and refining the object box in the corresponding bin. Most state-of-the-art detectors rely on anchors to enumerate the possible locations, scales, and aspect ratios for target objects (Liu et al., 2018). Anchors are regression references and classification candidates to predict proposals for two-stage detectors or final bounding boxes for single-stage detectors. Nevertheless, *anchors can be regarded as a feature-sharing sliding window scheme to cover the possible locations of objects*.

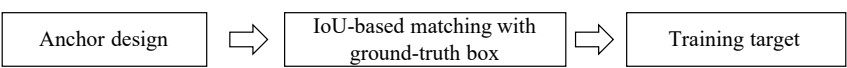

Figure 1: The anchor-based object detection frameworks need to (a) design anchors according to the ground-truth box distributions; (b) match anchors with ground-truth boxes to generate training target (anchor classification and refinement); and (c) utilize the target generated by (b) for training.

However, anchors must be carefully designed and used in object detection frameworks. (a) One of the most important factors in designing anchors is how densely it covers the instance location space. To achieve a good recall rate, anchors are carefully designed based on the statistics computed from the training/validation set (Lin et al., 2018). (b) One design choice based on a particular dataset is not always applicable to other applications, which harms the generality (Yang et al., 2018). (c) At training phase, anchor-methods rely on the *intersection-over-union* (IoU) to define the positive/negative samples, which introduces additional computation and hyper-parameters for an object detection system (Wang et al., 2019).

In contrast, our human vision system can recognize the instance in space and predict the boundary given the visual cortex map, without any pre-defined shape template (Bear et al., 2007). In other words, we human naturally recognize the object in the visual scene without enumerating the candidate boxes. Inspired by this, an intuitive question to ask is, *is the anchor scheme the optimal way to guide the search of objects*? And further, *could we design an accurate object detection framework without anchors or candidate boxes*? Without anchors, one may expect a complex method is required to achieve comparable performance. However, we show that a surprisingly simple and flexible system can match, even surpass the prior state-of-the-art object detection results without any requirement of candidate boxes.

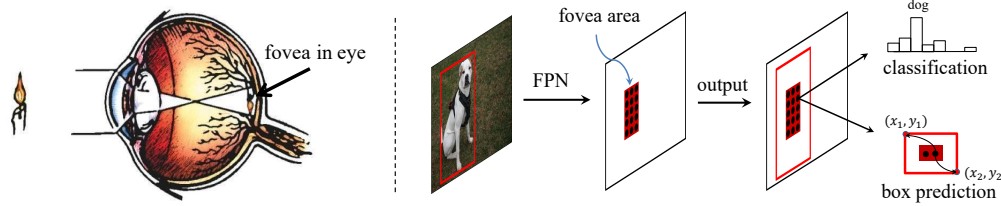

Figure 2: FoveaBox object detector. For each output spacial position that potentially presents an object, FoveaBox directly predicts the confidences for all target categories and the bounding box.

To this end, we present FoveaBox, a completely anchor-free framework for object detection. FoveaBox is motivated from the fovea of human eyes: the center of the vision field is with the highest visual acuity (Fig.2 left), which is necessary for activities where visual detail is of primary importance (Iwasaki & Inomata, 1986). FoveaBox jointly predicts the locations where the object's center area is likely to exist as well as the bounding box at each valid location.

In FoveaBox, each target object is predicted by category scores at center area, associated with 4-d bounding box, as shown in Fig.2 right. At training phase, we do not need to utilize anchors, or IoU matching to generate training target. Instead, the training target is directly generated by ground-truth boxes.

To demonstrate the effectiveness of the proposed detection scheme, we combine the recent progress of feature pyramid networks and our detection head to form the framework of FoveaBox. Without bells and whistles, FoveaBox gets state-of-the-art single-model results on the COCO object detection task. Compared with the anchor-based RetinaNet, FoveaBox gets 2.2 AP gains, which also surpasses most of previously published anchor based single-model results. We believe the simple training/inference manner of FoveaBox, together with the flexibility and accuracy, will benefit future research on object detection and relevant topics.

## 2 FOVEABOX

FoveaBox is conceptually simple: It contains a backbone network and a fovea head network. The backbone is responsible for computing a convolutional feature map over an entire input image and is an off-the-shelf convolutional network. The fovea head is composed of two sub-branches, the first branch performs per pixel classification on the backbone's output; the second branch performs box prediction for each position that potentially covered by an object.

## 2.1 REVIEW OF FPN AND ANCHORS:

We begin by briefly reviewing the Feature Pyramid Network (FPN) used for object detection (Lin et al., 2017). In general, FPN uses a top-down architecture with lateral connections to build an in-network feature pyramid from a single-scale input. FPN is independent of a specific task. For object detection, each level of the pyramid in FPN is used for detecting objects at a specific scale. On each feature pyramid, anchor-based methods uniformly place $A$ anchors on each of the $H \times W$ spacial position. After computing the IoU overlap between all anchors and the ground-truth boxes, the anchor-based methods can define training targets. Finally, the pyramid features are utilized to optimize the targets.

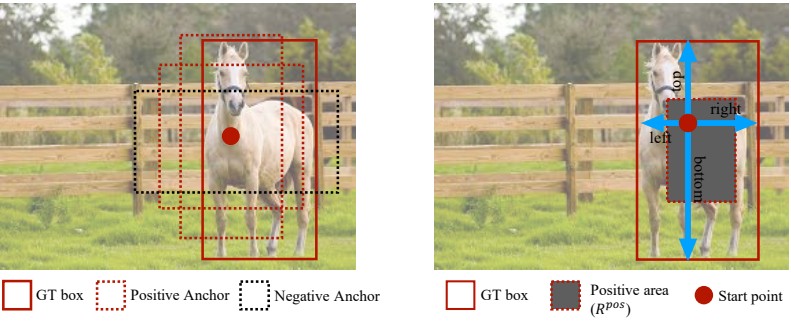

Figure 3: Anchor-based object detection v.s. FoveaBox object detection. **left:** The anchor-based method uniformly places $A$ ($A = 3$ in this example) anchors on each output spacial position, and utilizes IoU to define the positive/negative anchors; **right:** FoveaBox directly define positive/negative samples for each output spacial position by ground-truth boxes, and predicts the box boundaries from the corresponding position.

## 2.2 FOVEABOX

FoveaBox directly predicts the object existing possibility and the corresponding boundary for each position potential contained by an instance. In this section, we introduce the key components step-by-step.

### 2.2.1 OBJECT OCCURRENCE POSSIBILITY

Given a valid ground-truth box denoted as $(x_1, y_1, x_2, y_2)$. We first map the box into the target feature pyramid $P_l$

$$x_1' = \frac{x_1}{s_l}, \ y_1' = \frac{y_1}{s_l}, \ x_2' = \frac{x_2}{s_l}, \ y_2' = \frac{y_2}{s_l},$$
$$c_x' = 0.5(x_2' + x_1'), \ c_y' = 0.5(y_2' + y_1'), \ w' = x_2' - x_1', \ h' = y_2' - y_1', \tag{1}$$

where $s_l$ is the down-sample factor. The positive area $R^{pos}$ on the score map is designed to be roughly a shrunk version of the original one (Fig.3 right):

$$x_1^{pos} = c_x' - 0.5\sigma w', \ y_1^{pos} = c_y' - 0.5\sigma h',$$
$$x_2^{pos} = c_x' + 0.5\sigma w', \ y_2^{pos} = c_y' + 0.5\sigma h', \tag{2}$$

where $\sigma$ is the shrunk factor. At training phase, each cell inside the positive area is annotated with the corresponding target class label. The negative area is the whole feature map excluding area in $R^{pos}$. For predicting, each output set of pyramidal heat-map has $C$ channels, where $C$ is the number of categories, and is of size $H \times W$. Each channel is a binary mask indicating the possibility for a class, like FCNs in semantic segmentation (Long et al., 2015). The positive area usually accounts for a small portion of the whole feature map, so we adopt Focal Loss (Lin et al., 2018) to train this branch.

### 2.2.2 SCALE ASSIGNMENT

While our goal is to predict the boundary of the target objects, directly predicting these numbers is not stable, due to the large scale variations of the objects. Instead, we divide the scales of objects into several bins, according to the number of feature pyramidal levels. Each pyramid has a basic scale $r_l$ ranging from $32$ to $512$ on pyramid levels $P_3$ to $P_7$, respectively. The valid scale range of the target boxes for pyramid level $l$ is computed as

$$[r_l/\eta, r_l \cdot \eta], \tag{3}$$

where $\eta$ is set empirically to control the scale range for each pyramid. Target objects not in the corresponding scale range are ignored during training. Note that an object may be detected by multiple pyramids of the networks, which is different from previous practice that maps objects to only one feature pyramid (He et al., 2017).

### 2.2.3 BOX PREDICTION

Each ground-truth bounding box is specified in the way $G = (x_1, y_1, x_2, y_2)$. Starting from a positive point $(x, y)$ in $R^{pos}$, FoveaBox directly computes the normalized offset between $(x, y)$ and four boundaries:

$$
\begin{aligned}
t_{x_1} &= \log \frac{s_l(x + 0.5) - x_1}{r_l}, \quad t_{y_1} = \log \frac{s_l(y + 0.5) - y_1}{r_l}, \\
t_{x_2} &= \log \frac{x_2 - s_l(x + 0.5)}{r_l}, \quad t_{y_2} = \log \frac{y_2 - s_l(y + 0.5)}{r_l}.
\end{aligned}
\tag{4}
$$

This function first maps the coordinate $(x, y)$ to the input image, then computes the normalized offset between the projected coordinate and $G$. Finally the targets are regularized with the log-space function. $r_l$ is the basic scale defined in section 2.2.2.

For simplicity, we adopt the widely used Smooth $L_1$ loss (Ren et al., 2015) to train the box prediction $L_{box}$. After targets being optimized, we can generate the box boundary for each cell $(x, y)$ on the output feature maps[1]. In box branch, each output set of pyramidal heatmap has 4 channels, for jointly prediction of $(t_{x_1}, t_{y_1}, t_{x_2}, t_{y_2})$.

### 2.2.4 NETWORK ARCHITECTURE

To demonstrate the generality of our approach, we instantiate FoveaBox with multiple architectures. For clarity, we differentiate between: (i) the convolutional *backbone* architecture used for feature extraction over an entire image, and (ii) the network *head* for computing the final results.

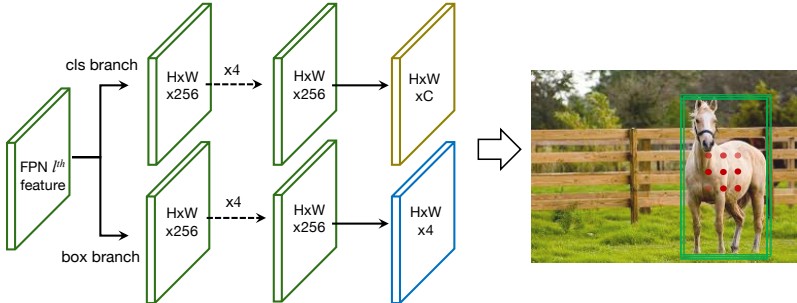

Figure 4: On each FPN feature level, FoveaBox attaches two subnetworks, one for classifying the corresponding cells and one for predict the $(x_1, y_1, x_2, y_2)$ of ground-truth object box. Right is the score output map with their corresponding predicted boxes before feeding into non-maximum suppression (NMS). The score probability in each position is denoted by the color density. More examples are shown in Fig.5.

---

[1]Eq.(4) and its inverse transformation can be easily implemented by an element-wise layer in modern deep learning frameworks (Paszke et al., 2017; Chen et al., 2016).

Most of the experiments are based on the head architecture as shown in Fig.4. We also utilize different head variants to further study the generality. More complex designs have the potential to improve performance but are not the focus of this work.

### 2.2.5 Implementations

We adopt the widely used FPN networks for fair comparison. Concretely, we construct a pyramid with levels $\{P_l\}, l = 3, 4, \cdots, 7$, where $l$ indicates pyramid level. $P_l$ has $1/2^l$ resolution of the input. All pyramid levels have $C = 256$ channels. Fovea head is attached on each pyramid level. Parameters are shared across all pyramid levels.

FoveaBox is trained with stochastic gradient descent (SGD). We use synchronized SGD over 4 GPUs with a total of 16 images per minibatch (4 images per GPU). Unless otherwise specified, all models are trained for 12 epochs with an initial learning rate of 0.01, which is then divided by 10 at 8th and again at 11th epochs. Weight decay of 0.0001 and momentum of 0.9 are used. Only standard horizontal image flipping is used for data augmentation. During inference, we first use a confidence threshold of 0.05 to filter out predictions with low confidence. Then, we select the top 1000 scoring boxes from each prediction layer. Next, NMS with threshold 0.5 is applied for each class separately. Finally, the top-100 scoring predictions are selected for each image. Although there are more intelligent ways to perform post-processing, such as bbox voting (Gidaris & Komodakis, 2015), Soft-NMS (Bodla et al., 2017) or test-time image augmentations, in order to keep simplicity and to fairly compare against the baseline models, we do not use those tricks here.

## 3 Experiments

We present experimental results on the bounding box detection track of the MS COCO benchmark. All models are trained on MS COCO `trainval35k`. If not specified, ResNet-50-FPN backbone and a 600 pixel train and test image scale are used to do the ablation study. We report lesion and sensitivity studies by evaluating on the `minival` split. For our main results, we report COCO AP on the `test-dev` split, which has no public labels and requires use of the evaluation server.

### 3.1 Ablation Study

**Various anchor densities and FoveaBox:** One of the most important design factors in an anchor-based detection system is how densely it covers the space of possible objects. As anchor-based detectors use a fixed sampling grid, a popular approach for achieving high coverage of boxes is to use multiple anchors at each spatial position. One may expect that we can always get better performance when attaching denser anchors on each position. To verify this assumption, we sweep over the number of scale and aspect ratio anchors used at each spatial position and each pyramid level in RetinaNet, including a single square anchor at each location to 12 anchors per location (Table.1(a)). Increasing beyond 6-9 anchors does not show further gains. The saturation of performance *w.r.t.* density implies the handcrafted, over-density anchors do not offer an advantage.

Over-density anchors not only increase the foreground-background optimization difficulty, but also likely to cause the *ambiguous* position definition problem. For each output spatial location, there are $A$ anchors whose labels are defined by the IoU with the ground-truth. Among them, some of the anchors are defined as positive samples, while others are negatives. However they are sharing the same input features. *The classifier needs to not only distinguish the samples from different positions, but also different anchors at the same position*.

In contrast, FoveaBox explicitly predicts one target at each position and gets no worse performance than the best anchor-based model. Compare with the anchor based scheme, FoveaBox enjoys several advantages. (a) Since we only predict one target at each position, the output space has been reduced to $1/A$ of the anchor-based method. (b) There is no ambiguous problem and the optimization target is more straightforward. (c) FoveaBox has fewer hyper-parameters, and is more flexible, since we do not need to extensively design anchors to see a relatively better choice.

**FoveaBox is more robust to box distribution:** One of the major benefits of FoveaBox is the robust prediction of bounding boxes. To verify this, we divide the boxes in the validation set into three groups according to the ground-truth aspect ratios $u = \max(\frac{h}{w}, \frac{w}{h})$. We compare FoveaBox and

Table 1: Ablation experiments for FoveaBox. All models are trained on `trainval35k`, test on `minival`. If not specified, default values are $\eta = 2.0$ and $\sigma = 0.4$. (a) Our anchor-free FoveaBox get 0.9 AP gains compared with the best model of anchor-based RetinaNet; (b) FoveaBox is more robust to bounding box distributions; (c) FoveaBox could also generate high-quality region proposals; (d) Accuracy of FoveaBox for various network depths and image scales; (e) and (f): FoveaBox gets best performance with $\eta = 2.0$ and $\sigma = 0.4$. See Section 3.1 for details.

(a) **Varying anchor density** and FoveaBox.

| method | #sc | #ar | AP | $AP_{50}$ | $AP_{75}$ |
|---|---|---|---|---|---|
| RetinaNet | 1 | 1 | 30.2 | 49.0 | 31.7 |
| RetinaNet | 2 | 1 | 31.9 | 50.0 | 34.1 |
| RetinaNet | 3 | 1 | 31.9 | 49.4 | 33.8 |
| RetinaNet | 2 | 3 | 34.2 | 53.1 | 36.5 |
| RetinaNet | 3 | 3 | 34.2 | 53.2 | 36.9 |
| RetinaNet | 4 | 3 | 33.9 | 52.1 | 36.2 |
| FoveaBox | - | - | **35.1** | **54.3** | **37.1** |

(b) Detection with **different aspect ratios**

| method | AP | $AP_{u<3}$ | $AP_{3 \leq u < 5}$ | $AP_{u \geq 5}$ |
|---|---|---|---|---|
| RetinaNet | 34.2 | 36.5 | 24.5 | 10.2 |
| FoveaBox | **35.1** | **36.8** | **26.8** | **16.4** |

(c) **Region proposal** performance.

| method | backbone | $AR_{100}$ | $AR_{300}$ | $AR_{1000}$ |
|---|---|---|---|---|
| RPN | ResNet-50 | 44.5 | 51.1 | 56.6 |
| FoveaBox | ResNet-50 | **52.9** | **57.3** | **61.5** |

(d) Different input resolutions and models.

| net-depth-scale | AP | $AP_{50}$ | $AP_{75}$ |
|---|---|---|---|
| FoveaBox-50-400 | $31.9_{+1.4}$ | 49.6 | 33.8 |
| FoveaBox-50-600 | $35.1_{+0.9}$ | 54.3 | 37.1 |
| FoveaBox-50-800 | $36.4_{+0.9}$ | 56.2 | 38.7 |
| FoveaBox-101-400 | $33.3_{+1.4}$ | 51.0 | 35.0 |
| FoveaBox-101-600 | $37.0_{+1.0}$ | 56.4 | 39.3 |
| FoveaBox-101-800 | $38.6_{+0.9}$ | 58.0 | 41.2 |

(e) Varying $\eta$ ($\sigma = 0.4$).

| $\eta$ | AP | $AP_{50}$ | $AP_{75}$ |
|---|---|---|---|
| 1.0 | 32.0 | 50.4 | 31.8 |
| 1.5 | 34.1 | 53.3 | 36.0 |
| 2.0 | **35.1** | **54.4** | **37.0** |
| 2.5 | 35.0 | 54.2 | 36.8 |
| 3.0 | 34.3 | 53.3 | 36.5 |
| 4.0 | 32.8 | 51.0 | 34.5 |

(f) Varying $\sigma$ ($\eta = 2.0$).

| $\sigma$ | AP | $AP_{50}$ | $AP_{75}$ |
|---|---|---|---|
| 0.2 | 34.1 | 53.2 | 36.0 |
| 0.3 | 34.8 | 54.0 | 36.7 |
| 0.4 | **35.1** | **54.4** | **37.0** |
| 0.5 | 34.8 | 53.9 | 36.6 |
| 0.6 | 34.1 | 53.1 | 36.0 |
| 0.7 | 33.3 | 52.5 | 34.9 |

RetinaNet at different aspect ratio thresholds, as shown in Table 1(b). We see that both methods get best performance when $u$ is low. Although FoveaBox also suffers performance decrease when $u$ increases, it is much better than the baseline model.

**Generating high-quality region proposals:** Changing the classification target to class-agnostic head is straightforward and could generate region proposals. We compare the proposal performance against FPN-based RPN (Lin et al., 2017) and evaluate average recalls (AR) with different numbers of proposals on `minival` set, as shown in Table 1(c). Surprisingly, our method outperforms the RPN baseline by a large margin, among all criteria. Specifically, with top 100 region proposals, FoveaBox gets 52.9 AR, outperforming RPN by 8.4 points. This validates that our model's capacity in generating high quality region proposals.

**Across model depth and scale:** Table 1(d) shows FoveaBox utilizing different backbone networks and input resolutions. The train/inference settings are exactly the same as the baseline method (Lin et al., 2018). Under the same settings, FoveaBox consistently gets 0.9~1.4 *higher AP*. When comparing the inference speed, we find that FoveaBox models are about 1.1~1.3 times *faster* than the RetinaNet counterparts.

**Analysis of $\eta$ and $\sigma$:** In Eq.(3), $\eta$ controls the scale assignment extent for each pyramid. As $\eta$ increases, each pyramid will response to more scales of objects. Table 1(e) shows the impact of $\eta$ on the final detection performance. Another important hyper-parameter is the shrunk factor $\sigma$ which controls the positive/negative samples. Table 1(f) shows the model performance with respect to $\sigma$ changes. In this paper, $\sigma = 0.4$ and $\eta = 2$ are used in other experiments.

**IoU-based assignment v.s. fovea area:** Another choice of defining the positive/negative samples is firstly gets the predicted box from the box branch, and then assign the target labels based on the IoU between the predicted boxes and ground-truth boxes. As shown in Table 2, the shrunk version gets better performance (+0.4 AP) than the IoU-based assignment process.

**Better head and feature alignment:** The most recent works (Chen et al., 2019; Yang et al., 2019) suggest to align the features in one-stage object detection frameworks with anchors. In FoveaBox, we adopt deformable convolution (Dai et al., 2017) based on the box offset learned by Eq.(4) to refine the classification branch[2]. FoveaBox works well when adding such techniques. Specifically,

---

[2]See supplementary for network details.

Table 2: Label assignment strategy (ResNet-50, 800 scale).

| assign method | AP | $AP_{50}$ | $AP_{75}$ |
|---|---|---|---|
| IoU (0.5/0.4) | 35.6 | 54.7 | 37.7 |
| IoU (0.6/0.5) | 35.9 | 54.7 | 38.4 |
| IoU (0.5/0.5) | 36.0 | 54.9 | 38.4 |
| Fovea ($\sigma = 0.4$) | 36.4 | 56.2 | 38.7 |

when we change the classification branch to a heavier head, together with feature alignment and GN, FoveaBox gets 40.1 AP using ResNet-50 as backbone! This experiment demonstrates the generality of our approach to the network design (Table 3).

Table 3: Feature alignment and group normalization (ResNet-50, 800 scale).

| cls branch | alignment | GN | AP | $AP_{50}$ | $AP_{75}$ |
|---|---|---|---|---|---|
| $256_{(3\times3)} \rightarrow 256_{(3\times3)} \rightarrow 256_{(3\times3)} \rightarrow 256_{(3\times3)}$ | | | 36.4 | 56.2 | 38.7 |
| $256_{(3\times3)} \rightarrow 256_{(3\times3)} \rightarrow 256_{(3\times3)} \rightarrow 256_{(3\times3)}$ | ✓ | | 36.8 | 56.5 | 38.9 |
| $256_{(3\times3)} \rightarrow 256_{(3\times3)} \rightarrow 256_{(3\times3)} \rightarrow 256_{(3\times3)}$ | ✓ | ✓ | 37.1 | 56.7 | 39.2 |
| $1024_{(3\times3)} \rightarrow 1024_{(1\times1)}$ | | | 36.7 | 57.0 | 39.1 |
| $1024_{(3\times3)} \rightarrow 1024_{(1\times1)}$ | ✓ | | 37.2 | 57.4 | 39.4 |
| $1024_{(3\times3)} \rightarrow 1024_{(1\times1)}$ | ✓ | ✓ | 37.5 | 58.2 | 39.5 |
| $1024_{(3\times3)} \rightarrow 1024_{(1\times1)}$, $2\times$ epochs | ✓ | ✓ | 37.9 | 58.4 | 40.4 |
| $1024_{(3\times3)} \rightarrow 1024_{(1\times1)}$, $2\times$ epochs, mstrain | ✓ | ✓ | 40.1 | 60.8 | 42.5 |

## 3.2 MAIN RESULTS

We compare FoveaBox to the state-of-the-art methods in Table 4. All instantiations of our model outperform baseline variants of previous state-of-the-art models. The first group of detectors on Table 4 are two-stage detectors, the second group one-stage detectors, and the last group the FoveaBox detector. FoveaBox outperforms all single-stage detectors under ResNet-101 backbone, under all evaluation metrics. This includes the recent one-stage CornerNet and ExtremeNet (Law & Deng, 2018; Zhou et al., 2019b). FoveaBox also outperforms most of two-stage detectors, including FPN (Lin et al., 2017), Mask R-CNN (He et al., 2017) and IoU-Net (Jiang et al., 2018).

Two-stage detectors rely on region-wise sub-networks to further classify the sparse region proposals. Since FoveaBox could also generate region proposals by changing the model head to class agnostic scheme (Table 1(c)), we believe it could further improve the performance of two-stage detectors, which beyond the focus of this paper.

## 4 IN CONTEXT OF RELATED WORK

Our work is related to previous works in different aspects. Before closing, we discuss the relations and differences in details.

**Anchor-based Object Detection**: The anchor-based object detection frameworks can be generally grouped into two factions: two-stage, proposal driven detectors and one-stage, proposal free methods. Anchors are regression references and classification candidates to predict proposals for two-stage detectors(Ren et al., 2015; Lin et al., 2017; He et al., 2017; Cai & Vasconcelos, 2018) or final bounding boxes for single-stage detectors (Liu et al., 2016; Lin et al., 2018; Redmon & Farhadi, 2017). Most top one-stage detectors rely on the anchor boxes to enumerate the possible locations of target objects.

**Anchor-Free Explorations**: There are also some prior works trying to remove the dependence of anchors. Due to the absence of anchors or region proposals, usually they lack the ability to deal with complex scenes and cases (Huang et al., 2015; Redmon et al., 2016). In text detection, the score mask technique has been used due to the arbitrary shape of target text (Zhang et al., 2016; Hu et al., 2017; Zhou et al., 2017). Such works usually utilize the fully convolutional networks to predict the

Table 4: **Object detection** single-model results *v.s.* state-of-the-arts on COCO `test-dev`. We show results for our FoveaBox models with 800 input scale. FoveaBox-align indicates utilizing feature alignment discussed in Section 3.1.

| | backbone | AP | $AP_{50}$ | $AP_{75}$ | $AP_S$ | $AP_M$ | $AP_L$ |
|---|---|---|---|---|---|---|---|
| *two-stage methods* | | | | | | | |
| Faster R-CNN w FPN (Lin et al., 2017) | ResNet-101 | 36.2 | 59.1 | 39.0 | 18.2 | 39.0 | 48.2 |
| Mask R-CNN (He et al., 2017) | ResNet-101 | 38.2 | 60.3 | 41.7 | 20.1 | 41.1 | 50.2 |
| Faster R-CNN by G-RMI (Huang et al., 2017b) | Inception-ResNet-v2 | 34.7 | 55.5 | 36.7 | 13.5 | 38.1 | 52.0 |
| Faster R-CNN w TDM (Shrivastava et al., 2016) | Inception-ResNet-v2 | 36.8 | 57.7 | 39.2 | 16.2 | 39.8 | 52.1 |
| Relation Network (Hu et al., 2018) | DCN-101 | 39.0 | 58.6 | 42.9 | - | - | - |
| IoU-Net (Jiang et al., 2018) | ResNet-101 | 40.6 | 59.0 | - | - | - | - |
| Cascade R-CNN (Cai & Vasconcelos, 2018) | ResNet-101 | 42.8 | 62.1 | 46.3 | 23.7 | 45.5 | 55.2 |
| *one-stage methods* | | | | | | | |
| YOLOv2 (Redmon et al., 2016) | Darknet-19 | 21.6 | 44.0 | 19.2 | 5.0 | 22.4 | 35.5 |
| YOLOv3 (Redmon & Farhadi, 2018) | Darknet-53 | 33.0 | 57.9 | 34.4 | 18.3 | 35.4 | 41.9 |
| SSD513 (Fu et al., 2017) | ResNet-101 | 31.2 | 50.4 | 33.3 | 10.2 | 34.5 | 49.8 |
| DSSD513 (Fu et al., 2017) | ResNet-101 | 33.2 | 53.3 | 35.2 | 13.0 | 35.4 | 51.1 |
| RetinaNet (Lin et al., 2018) | ResNet-101 | 39.1 | 59.1 | 42.3 | 21.8 | 42.7 | 50.2 |
| RetinaNet (Lin et al., 2018) | ResNeXt-101 | 40.8 | 61.1 | 44.1 | 24.1 | 44.2 | 51.2 |
| RPDet (Yang et al., 2019) | ResNeXt-101 | 41.0 | 62.9 | 44.3 | 23.6 | 44.1 | 51.7 |
| FCOS (Tian et al., 2019) | ResNeXt-101 | 42.1 | 62.1 | 45.2 | 25.6 | 44.9 | 52.0 |
| CornerNet (Law & Deng, 2018) | Hourglass-104 | 40.5 | 56.5 | 43.1 | 19.4 | 42.7 | 53.9 |
| ExtremeNet (Zhou et al., 2019b) | Hourglass-104 | 40.1 | 55.3 | 43.2 | 20.3 | 43.2 | 53.1 |
| CenterNet (Duan et al., 2019) | Hourglass-104 | 42.1 | 61.1 | 45.9 | 24.1 | 45.5 | 52.8 |
| *ours* | | | | | | | |
| FoveaBox | ResNet-101 | 40.8 | 61.4 | 44.0 | 24.1 | 45.3 | 53.2 |
| FoveaBox | ResNeXt-101 | 42.3 | 62.9 | 45.4 | 25.3 | 46.8 | 55.0 |
| FoveaBox-align | ResNet-101 | 42.1 | 62.7 | 45.5 | 25.2 | 46.6 | 54.5 |
| FoveaBox-align | ResNeXt-101 | **43.9** | **63.5** | **47.7** | **26.8** | **46.9** | **55.6** |

existence of target scene text and the quadrilateral shapes. Guided-Anchoring (Wang et al., 2019) jointly predicts the locations where the center of objects are likely to exist as well as the scales and aspect ratios centered at the corresponding locations. Guided-Anchoring still relies on predefined anchors to optimize the object shape, and utilizes the center points to give the best predictions. In contrast, FoveaBox predicts the (*left, top, right, bottom*) boundaries of the object for each foreground position.

**Contemporary Works**: Also there are contemporary works (Tian et al., 2019; Zhou et al., 2019a) similar to the idea of FoveaBox. FCOS relies on the proposed centerness map for better learning of the instance. Instead, FoveaBox directly predict the final class probability without centerness voting, which is more simple. The CenterNet (Zhou et al., 2019a) represent each instance by its features at the center point, which could also get comparable performance when adopting heavier networks (Law & Deng, 2018). Compare with a single point, our fovea-based positive sample definition process is more reasonable (The performance drops dramatically when we decrease $\sigma$). Again, we note that FoveaBox, CenterNet and FCOS are *concurrent* works.

**Bottom-up Methods**: In CornerNet (Law & Deng, 2018), the authors propose to detect an object bounding box as a pair of key-points, the top-left corner and the bottom-right corner. CornerNet adopts the Associative Embedding (Newell et al., 2017) technique to separate different instances. Also there are some following works in bottom-up grouping manner (Zhou et al., 2019b; Duan et al., 2019). It should be noted that the bottom-up methods also do not need anchors during training and inference.

## 5 CONCLUSION

We have presented FoveaBox, a simple, effective, and completely anchor-free framework for generic object detection. By simultaneously predict the object position and the corresponding boundary, FoveaBox gives a clean solution for detecting objects without prior candidate boxes. We demonstrate its effectiveness on standard benchmarks and report extensive experimental analysis. We believe the simple and effective approach will serve as a solid baseline and help ease future research for object detection.

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

# A APPENDIX

## A.1 QUALITATIVE RESULTS

Fig.5 shows the detection outputs of FoveaBox. Points and boxes with class probability larger than 0.5 are shown (before feeding into NMS). For each object, though there are several active points, the predicted boxes are very close to the ground-truth. These figures demonstrate that FoveaBox could directly generate accurate, robust box predictions, without the requirement of candidate anchors.

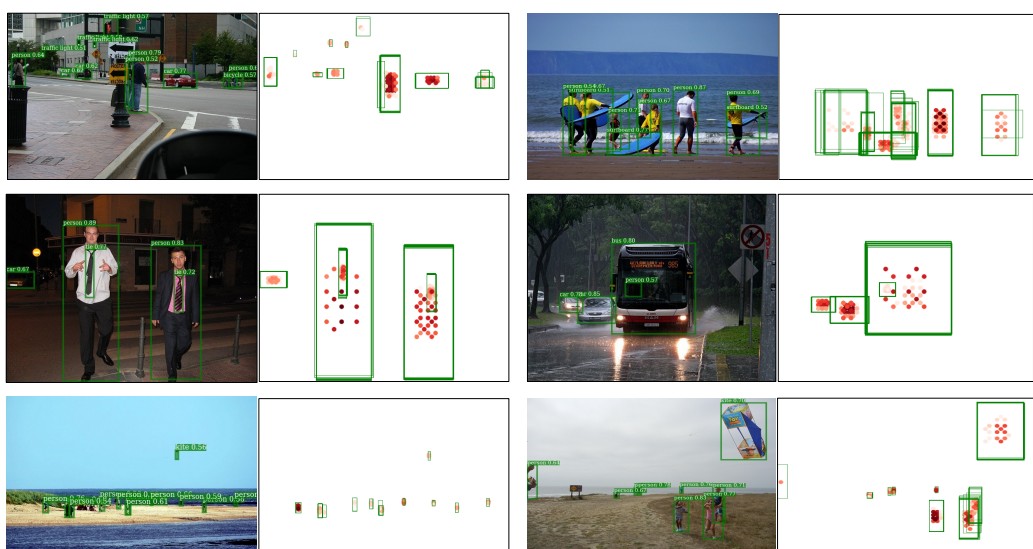

Figure 5: **FoveaBox results on the COCO `minival` set**. These results are based on ResNet-101, achieving a single model box AP of 38.6. For each pair, left is the detection results with bounding box, category, and confidence. Right is the score output map with their corresponding bounding boxes before feeding into non-maximum suppression (NMS). The score probability in each position is denoted by the color density.

## A.2 PER-CLASS DIFFERENCE:

Fig. 6 shows per-class AP difference of FoveaBox and RetinaNet. Both of them are with ResNet-50-FPN backbone and 800 input scale. The vertical axis shows $AP_{FoveaBox}$-$AP_{RetinaNet}$. FoveaBox shows improvement in most of the classes.

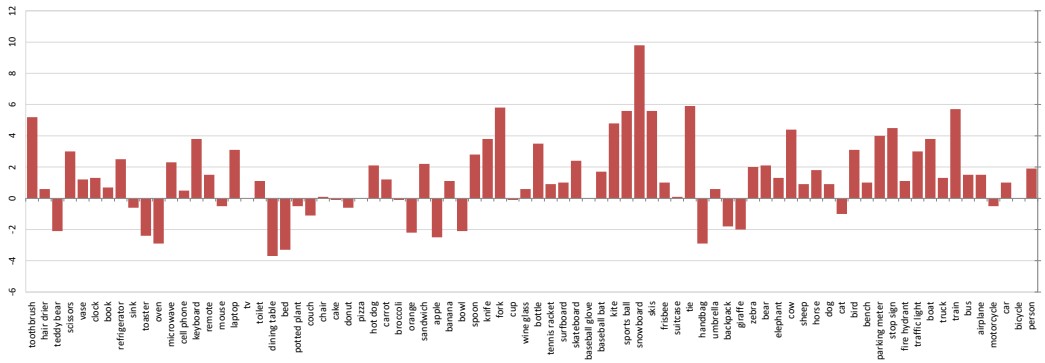

Figure 6: AP difference of FoveaNet and RetinaNet on COCO dataset. Both models use ResNet-FPN-50 as backbone and 800 input scales.

## A.3 MORE TRAINING TIME:

Table 5 shows the detection performance under 2x setting (24 epochs training). FoveaBox also outperforms the anchor-based baseline method.

Table 5: 2x epochs training (800 scale).

| method | AP | $AP_{50}$ | $AP_{75}$ |
|---|---|---|---|
| RetinaNet-50 | 36.4 | 56.3 | 39.4 |
| RetinaNet-101 | 38.1 | 57.8 | 41.1 |
| FoveaBox-50 | 37.1 | 56.7 | 39.3 |
| FoveaBox-101 | 38.7 | 58.4 | 41.1 |

## A.4 FEATURE ALIGNMENT

For feature alignment, we adopt a 3×3 deformable convolutional layer to implement the transformation, as shown in Fig.7. The offset input to the deformable convolutional layer is $(\hat{t}_{x_1}, \hat{t}_{y_1}, \hat{t}_{x_2}, \hat{t}_{y_2})$ of the bbox output.

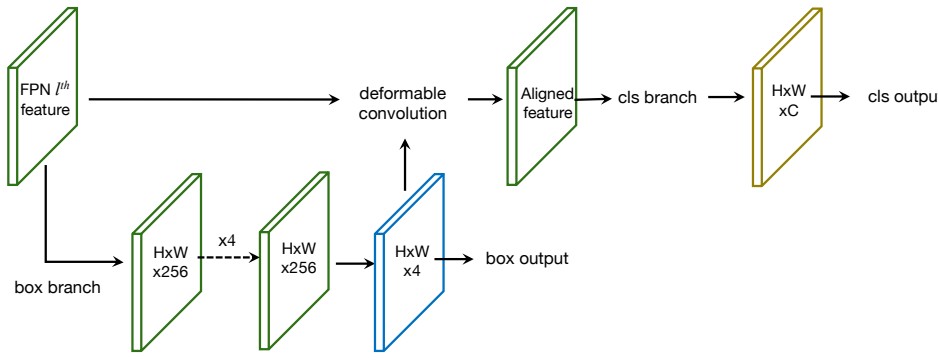

Figure 7: Feature alignment process.

