# OpenReview forum: "FoveaBox: Beyound Anchor-based Object Detection"
_ICLR.cc/2020/Conference — Reject_

### Official Review · AnonReviewer2 · 2019-10-26
**Official Blind Review #2**

**Rating:** 6

**Review:**

## Overview
The paper tackles object detection without predefined anchors (sliding windows).
The paper is well motivated and existing detection methods often rely on anchors, which may limits their potential. So this paper is solving an interesting problem and seems novel.
The paper provides experiments to support the proposed architecture empirically.

## Summary of the contribution:
1. The paper proposed an object detection approach called FoveaBox that does not rely on anchors (sliding widows).
2. The paper shows that FoveaBox outperforms some existing object detection methods.
3. The paper shows that FoveaBox can also be used for object proposals by changing the classification target to class-agnostic head.

## My feedback:
I think the paper is well motivated and has value to the object detection community. So I am leaning positive to accept this paper. The major reasons are:
1. The paper studies an interesting architecture which does not rely on sliding windows and shows its effectiveness. The problem seems not very well studied in the existing object detection literature.
2. The paper provides sufficient ablation study to analyze and understand the proposed methods.
However, I believe the paper could be significantly improved especially in the writing. So my position is not strong.

## Improvements
1. The writing can be significantly improved. The paper reads a bit confusing and unclear especially at the technical part. The paper could benefit from a clear overview figure about the proposed approach. Figure 4 seems to be doing that illustration but giving only the tensor shape seems quite confusing. I also listed some questions below about the actual algorithm.
2. The paper could be improved with experiments and ablation on another dataset. Current ablation shows that \eta should be 2.0 for best performance but without a second dataset it is hard to say this value is general. So one may have to tune this parameters in different datasets.
3. The scale is still discretized. So that is essentially anchors in the scale space. I wonder how could the approach applies to scale to?

## Questions
1. How would the proposed approach address the issue when there are multiple bounding boxes around the same pixels? It seems the current approach is predicting a box per pixel?
2. What is the inference computation cost? And do you have a comparison with existing methods on the compute cost?

**Experience Assessment:**

I have published in this field for several years.

**Review Assessment: Checking Correctness Of Derivations And Theory:**

I assessed the sensibility of the derivations and theory.

**Review Assessment: Checking Correctness Of Experiments:**

I assessed the sensibility of the experiments.

**Review Assessment: Thoroughness In Paper Reading:**

I read the paper at least twice and used my best judgement in assessing the paper.

---

> ### Author Response · Authors · 2019-11-11
> **Response to Review #2**
>
> Thank you for the detailed review. We appreciate your comments on the contributions of our work and your valuable suggestions to improve our paper.
>
> --About the paper writing and overview figure.
> Thanks for pointing out this, and we will further revise the paper, especially the overview figure, to make it more clear and self-standing.
>
> --Ablation on another dataset.
> Per the reviewer's suggestion, we conducted additional experiments on Pascal VOC object detection dataset (http://host.robots.ox.ac.uk/pascal/VOC/index.html).  This dataset covers 20 object categories, and the performance is measured by mean average precision (mAP) at IoU=0.5. All variants are trained on VOC2007 trainval and VOC2012 trainval, and tested on VOC2007 test dataset. Based on ResNet-50, we can compare the performances of FoveaBox and RetinaNet.
>
> Method      |    backbone      |    mAP@0.5    |    speed    |    hyper-params
> RetinaNet  |    ResNet-50     |        75.5           |    74 ms    |    9 anchors, IoU-pos=0.5, IoU-neg=0.4
> RetinaNet  |    ResNet-50     |        75.1           |    69 ms    |    6 anchors, IoU-pos=0.5, IoU-neg=0.4
> RetinaNet  |    ResNet-50     |        74.3           |    60 ms    |    3 anchors, IoU-pos=0.5, IoU-neg=0.4
> FoveaBox  |    ResNet-50     |        76.6           |    61 ms    |    $\eta$=2.0
> FoveaBox  |    ResNet-50     |        76.0           |    61 ms    |    $\eta$=1.5
> FoveaBox  |    ResNet-50     |        76.4           |    61 ms    |    $\eta$=2.5
>
> From this experiment, we can see that $\eta$ is robust across different datasets. FoveaBox also outperforms RetinaNet in Pascal VOC dataset (+1.1 mAP).
>
>
> --The scale is still discretized
> Following FPN, we detect different sizes of objects on different levels of feature maps. Unlike anchor-based detectors, which assign anchor boxes with different sizes to different feature levels, we directly limit the predicting range for  GT boxes. The multi-level prediction can help predicting objects of different scales.
>
> --Multiple bounding boxes around the same pixels?
> We believe your concern is the ambiguous samples due to the overlapping in GT boxes. Here we calculate the ratios of ambiguous samples to all positive samples on COCO train 2017 split. With FPN, only 3.2% positive samples are ambiguous samples. Note that it does not imply that there are 3.2% samples FoveaBox cannot work, since these locations are associated with GT boxes with minimal area.
> We also count the detected boxes produced by the ambiguous locations, and only 1.5% detected boxes come from these locations. As shown in our experiments, these minimal samples do not make FoveaBox inferior to anchor-based methods.
>
> --Inference computation cost.
> Per the reviewer's suggestion, we evaluate the inference time of these methods. The experiments are performed on a single Nvidia V100 GPU  by averaging 10 runs.
>
> Method                                          |    backbone           |    AP       |    Speed
> CenterNet (Zhou et al., 2019)    |    Hourglass-104   |    42.1     |    119 ms
> ExtremNet (Zhou et al., 2019)   |    Hourglass-104   |    40.1     |    301 ms
> CornerNet (Law et al., 2018)      |    Hourglass-104   |    40.5     |    291 ms
> RetinaNet                                      |    ResNeXt-101      |    40.8     |    106 ms
> FCOS                                              |    ResNeXt-101       |    42.1     |     98 ms
> FoveaBox                                      |    ResNeXt-101       |    42.3     |     95 ms
>
> Since most of the inference time are spent in the ResNet backbone, the acceleration by our method is not so big compared to RetinaNet. We will include these experimental results in the revised version.

---

### Official Review · AnonReviewer3 · 2019-10-26
**Official Blind Review #3**

**Rating:** 6

**Review:**

This paper introduces an anchor-free object detection framework that aims at simultaneously predicting the object position and the corresponding boundary. To achieve this, the proposed FoveaBox detector predicts category-sensitive semantic maps for the object existing possibility, and  produces category-agnostic bounding box for each position that is likely to contain an object. The scales of target boxes are associated with feature pyramid representations. Experiments are performed on MS COCO detection benchmark.

Pros:
The proposed approach is simple and is shown to avoid most computation and hyper-parameters related to anchor boxes. The paper is well written and easy to follow.
Cons:
The main issue with the paper is the main idea is similar to [1,2, 3, 4, 5]. For instance, CenterNet also represents each object instance by its features at the center point and achieves similar detection performance compared to the proposed detector. Further, no speed comparison with these approaches is provided in the paper. Without a fair speed comparison and with similar detection performance, it is difficult to fully assess the merits of the proposed approach. Though inference speed comparison is reported with RetineNet. However, a proper and detailed comparison with [1, 2, 3, 4, 5] is missing.

1: Xingyi Zhou, Dequan Wang, Philipp Krähenbühl: Objects as Points. CoRR abs/1904.07850 (2019).
2: Xingyi Zhou, Jiacheng Zhuo, Philipp Krähenbühl: Bottom-Up Object Detection by Grouping Extreme and Center Points. CVPR 2019.
3: Zhi Tian, Chunhua Shen, Hao Chen, Tong He: FCOS: Fully Convolutional One-Stage Object Detection. CoRR abs/1904.01355 (2019).
4: Hei Law and Jia Deng: Cornernet: Detecting objects as paired keypoints. ECCV 2018.
5: Kaiwen Duan, Song Bai, Lingxi Xie, Honggang Qi, Qingming Huang, Qi Tian: CenterNet: Keypoint Triplets for Object Detection. CoRR abs/1904.08189 (2019).



**Experience Assessment:**

I have published in this field for several years.

**Review Assessment: Checking Correctness Of Derivations And Theory:**

I assessed the sensibility of the derivations and theory.

**Review Assessment: Checking Correctness Of Experiments:**

I carefully checked the experiments.

**Review Assessment: Thoroughness In Paper Reading:**

I read the paper thoroughly.

---

> ### Public Comment · ~Zhi_Tian2 · 2019-11-07
> **Relation between Foveabox and FCOS**
>
> I am one of the authors of FCOS (Tian et al., 2019). Just for your information, Foveabox and FCOS are contemporary and independent works, so FCOS should not weaken the originality of this work. Thank you.

---

> > ### Author Response · Authors · 2019-11-08
> > **Thanks for your reply**
> >
> > We thank Zhi Tian for pointing out the relation between FCOS and FoveaBox.

---

> ### Author Response · Authors · 2019-11-12
> **Response to Review #3**
>
> Thank you for the detailed review. The main concerns of the reviewer are addressed separately below.
>
> --The main idea is similar to [1, 2, 3, 4, 5].
> The listed works can be grouped into two categories: bottom-up methods [2, 4, 5] and top-down methods [1, 3].
> (a) Bottom-up methods [2, 4, 5]. CornerNet and ExtremeNet are recently proposed one-stage object detectors. The key idea is to detect pairs of corner keypoints and groups them to form the final detected boxes. CornerNet requires much more complicated post-processing to group the pairs of corners belonging to the same instance. An extra associative embedding is learned for the purpose of grouping. The method in [5] is based on CornerNet, and tries to detect each object as a triplet, rather than a pair of keypoints. The authors also designed cascade corner pooling and center pooling to further enrich the corner information. ExtremeNet requires a combinatorial grouping stage after keypoint detection, which significantly slows down the algorithm. Our FoveaBox, on the other hand, simply predicts the center area per object and boundaries without the need for grouping or post-processing.
> (b) Top-down methods [1, 3]. Here we note that [1, 3] and FoveaBox are contemporary and independent works. In [1], the authors try to model an object as a single point and its bounding box. The detector uses keypoint estimation to find center points and regresses to boundaries. The idea of our FoveaBox is most similar to FCOS [3]. In FCOS, the authors propose to utilize the additional “centerness” branch to suppress low-quality predicted boxes.  In contrast, the occurrence area proposed in our work could effectively suppress most false positive predictions. We also thank Zhi Tian for pointing out the relation between FCOS and FoveaBox.
>
> --Speed comparison
> Per the reviewer's suggestion, we evaluate the inference time of these methods. The experiments are performed on a single Nvidia V100 GPU  by averaging 10 runs. The speed field could remotely reflect the actual runtime of a model due to the difference in implementations.
>
> Method                                          |    backbone           |     AP       |    Speed
> CenterNet (Zhou et al., 2019)    |    Hourglass-104   |     42.1     |    119 ms
> CenterNet (Duan et al., 2019)   |    Hourglass-104   |     44.9     |    329 ms
> ExtremNet (Zhou et al., 2019)   |    Hourglass-104   |     40.1     |    301 ms
> CornerNet (Law et al., 2018)     |    Hourglass-104    |    40.5     |    291 ms
> RetinaNet                                     |    ResNeXt-101       |    40.8     |    106 ms
> FCOS                                             |    ResNeXt-101       |    42.1     |     98 ms
> FoveaBox                                     |    ResNeXt-101       |    42.3     |     95 ms
> FoveaBox-align                           |    ResNeXt-101       |    43.9     |     138 ms
>
> We will include these discussions and results in the revised version. Finally, thanks and we are looking forward to your further valuable response.

---

### Official Review · AnonReviewer1 · 2019-10-30
**Official Blind Review #1**

**Rating:** 3

**Review:**

The paper introduces foveabox, a method that performs "keypoint" like object detection -- instead of "anchor" based detection (to be discussed later). The idea is simple: predict class labels for pixels that fall within (a reduced version) the GT boxes of the instance; and predict bounding box offsets for those positive pixels. The idea is built on top of the FPN backbone, where a set of feature maps (each representing a specific scale) are used to detect object boxes in multiple scales.  The method is mainly compared against RetinaNet (which is "anchor" driven), and also compared against other more recent methods (ExtremeNet, CenterNet, FCOS) etc.

+ The paper is quite well written and structured, the illustrations are also clear;
+ I have also read its previous version, and the new version has added a significant amount of work improving it -- e.g. added feature alignment and group norm;
+ I haven't fully checked the results section of other concurrent papers for full comparison, but the current results are among the state-of-the-art for one-stage detectors.

- I don't think the paper breaks away from the notion of "anchors". The current approach is in fact implicitly defining a *single* anchor for each feature map, and changing the assignment rule from IoU based to distance/scale based. I firmly believe that such modification can lead to concrete improvements -- for example recent work has shown that changing assignment rule can improve AP even with fewer anchors (TensorMask); but to say that the paper breaks away from the usage of anchors, it is a bit far for me. I think I will be way more convinced if the experiments are done on a single-scale feature map (anchor also works with single-scale but I am not sure without defining different anchors it can work just on C4 for example).
- Also because such assignment rule has been there before (e.g. DeepMask has used x, y, scale for assignment), and with other recent works (CenterNet, Objects as Keypoints), the contribution of this work is fairly limited

Therefore, I vote for reject.

**Experience Assessment:**

I have published one or two papers in this area.

**Review Assessment: Checking Correctness Of Derivations And Theory:**

N/A

**Review Assessment: Checking Correctness Of Experiments:**

I carefully checked the experiments.

**Review Assessment: Thoroughness In Paper Reading:**

N/A

---

> ### Author Response · Authors · 2019-11-08
> **Response to Review #1**
>
> We thank the reviewer for the helpful feedback and suggestions. The main concerns of the reviewer are addressed separately below.
>
> --Comparison with "single" anchor
> FoveaBox views locations as training samples instead of anchors, like FCNs for semantic segmentation. It directly predicts the object existing possibility and boundary without any anchors. Previous works usually utilize "discrete" anchors to enumerate possible scales/aspect ratios on each output position and further refine the anchors. We believe that our scheme is simpler and more flexible. As shown in Table 1(a), FoveaBox gives 4.9 AP gains compared with single-anchor based RetinaNet.
>
> --On single-scale feature map
> Per the reviewer's concern, we conducted the ablation on single feature map of ResNet-50 (C4). In this ablation, we compare the region proposal performance with RPN (both on C4/FPN of ResNet-50). Here we can see that FoveaBox-C4 gets comparable performance with RPN-C4 (51.0 v.s. 51.6), and FoveaBox-C4 is much better than RPN-C4 with single anchor (+17.1). Another interesting observation is that FoveaBox-C4 gets comparable performance with RPN-FPN with single anchor (51.0 v.s. 50.1).
>
> Method                       |     backbone   |  AR@1000
> RPN                             |     R-50-C4       |  51.6
> RPN-single-anchor   |     R-50-C4       |  33.9
> FoveaBox                   |     R-50-C4       |  51.0
> RPN                             |     R-50-FPN    |  56.6
> RPN-single-anchor   |     R-50-FPN    |  50.1
> FoveaBox                   |     R-50-FPN    |  61.5
>
>
> --Comparison with DeepMask
> (a) DeepMask is designed for object proposals while FoveaBox is developed for object detection.
> (b) DeepMask relies on predefined patches on each output spatial position to define positive/negative training samples, and the predefined patches are similar to anchors. On the other hand, FoveaBox generates positive/negative training samples directly from GT boxes.
> (c) At inference phase, the patch is used in DeepMask to generate object mask and box. In FoveaBox, the object box is directly learnt by the box branch.
> (d) As for performance, the box proposal recall of FoveaBox is much higher than that in DeepMask (44.6AR@1000).
>
> --Comparison with TensorMask and CenterNet
> Here we first note that TensorMask, CenterNet and FoveaBox are concurrent works. TensorMask tries to investigate the paradigm of dense instance segmentation. We agree that CenterNet (Zhou et al., 2019) and our FoveaBox share similar ideas to detect objects, however as Zhi (one author of FCOS) said, concurrent works should not weaken the originality of our work.
>
> Finally, the ablations and discussions will be updated to the revised paper. Thanks and we are looking forward to your further valuable response.

---

### Public Comment · ~Lei_Wu5 · 2019-10-10
**Questions about Table 4**

Hi,

Thanks for sharing your work with our community. The idea of scale assignment is inspiring.

I have some questions about Table 4. There seem to be some errors in several entries of Table 4. For example, the CenterNet (Duan et al., 2019) with Hourglass-104 achieves a single-model single scale AP of 44.9% instead of 42.1% as reported in this paper. The FCOS (Tian et al., 2019) achieves 42.7% AP and 43.2% AP with ResNeXt-101-32x8d and ResNeXt-101-64x4d respectively, both higher than the reported AP (42.1%) in this manuscript. And for RPDet (Yang et al., 2019), there is no result based on ResNeXt-101. The original paper presents a ResNet-101-DCN version of RPDet, which is the equivalence of the FoveaBox-align with ResNet-101. And RPDet using ResNet-101-DCN also outperforms FoveaBox-align using ResNet-101 (42.8% vs 42.1%).

Given the above facts, it may be too aggressive to claim that all instantiations of the proposed method outperform baseline variants of previous state-of-the-art models.

---

> ### Author Response · Authors · 2019-10-10
> **Thanks for your interest in our paper**
>
> There are several versions of these methods in arXiv, the latest versions of these methods may get better performance based on better training/testing strategies.
> (a) CenterNet(Duan et al.,2019), we have checked the paper and the results are mistakenly taken from another CenterNet (Zhou et al., 2019) (https://arxiv.org/pdf/1904.07850.pdf). Thanks and we will fix the issue in the next version. Here we note that CenterNet are using original and  flipped image for inference.
> (b) FCOS (Tian et al., 2019), the original FCOS gets 42.1% AP (https://arxiv.org/pdf/1904.01355v3.pdf). The latest FCOS could get better performance (+0.6) with centerness-regression and center-sampling.
> (c) RPDet (Yang et al., 2019), we believe ResNet-DCN version of RPDet is not equivalent with FoveaBox-align, because FoveaBox-align only add one layer of deformable conv. On the other hand, ResNet-DCN version of RPDet uses DCN backbone and DCN head. FoveaBox-ResNet101-align (42.1%) is the equivalence of RPDet-ResNet101 (41.0%).  We believe our FoveaBox could get better performance with DCN backbone.  We will fix the mistake "ResNeXt-101---->ResNet-101".
>
> Finally, thank you for your valuable comments and suggestions.

---

### Decision · Program_Chairs · 2019-12-19

**Decision:**

Reject

**Comment:**

The paper proposes a method for object detection by predicting category-specific object probability and category-agnostic bounding box coordinates for each position that's likely to contain an object. The proposed idea is interesting and the experimental results show improvement over RetinaNet and other baselines. However, in terms of weakness, (1) conceptually speaking it's unclear whether the proposed method is a big departure from the existing frameworks; and (2) although the authors are claiming SOTA performance, the proposed method seems to be worse than other existing/recent work. Some example references are listed below (more available here: https://paperswithcode.com/paper/foveabox-beyond-anchor-based-object-detector).

[1] Scale-Aware Trident Networks for Object Detection
https://arxiv.org/abs/1901.01892

[2] GCNet: Non-local Networks Meet Squeeze-Excitation Networks and Beyond
https://arxiv.org/abs/1904.11492

[3] CBNet: A Novel Composite Backbone Network Architecture for Object Detection
https://arxiv.org/abs/1909.03625

[4] EfficientDet: Scalable and Efficient Object Detection
https://arxiv.org/abs/1911.09070

References [3] and [4] are concurrent works so shouldn't be a ground of rejection per se, but the performance gap is quite large. Compared to [1] and [2] which have been on arxiv for a while (+5 months) the performance of the proposed method is still inferior. Despite considering that object detection is a very competitive field, the conceptual/technical novelty and overall practical significance seem limited for ICLR. For a future submission, I would suggest that a revision of this paper being reviewed in a computer vision conference, rather than ML conference.